

# Technical Note: Relating functional group measurements to carbon types for improved model-measurement comparisons of organic aerosol composition

Satoshi Takahama[1] and Giulia Ruggeri[1]

[1]ENAC/IIE Swiss Federal Institute of Technology Lausanne (EPFL), Switzerland

*Correspondence to:* Satoshi Takahama (satoshi.takahama@epfl.ch)

**Abstract.** Functional group (FG) analysis provides a means by which functionalization in organic aerosol can be attributed to the abundances of its underlying molecular structures. However, performing this attribution requires additional, unobserved details about the molecular mixture to provide constraints in the estimation process. To address this issue, we present an approach for conceptualizing FG measurements of organic aerosol in terms of its functionalized carbon atoms. This reformulation facilitates estimation of mass recovery and biases in popular carbon-centric metrics that describe the extent of functionalization (such as oxygen to carbon ratio, organic mass to organic carbon mass ratio, and mean carbon oxidation state) for any given set of molecules and FGs analyzed. Furthermore, this approach allows development of parameterizations to more precisely estimate the organic carbon content from measured FG abundance. We use simulated photooxidation products of $\alpha$-pinene secondary organic aerosol previously reported by Ruggeri et al., (*Atmos. Chem. Phys.*, 16, 4401–4422, 2016) and FG measurements by Fourier Transform Infrared (FT-IR) spectroscopy in chamber experiments by Sax et al. (*Aerosol Sci. Tech.*, 39, 822–830, 2005) to infer the relationships among molecular composition, FG composition, and metrics of organic aerosol functionalization. We find that for this simulated system, ∼80% of the carbon atoms should be detected by FGs for which calibration models are commonly developed, and ∼7% of the carbon atoms are undetectable by FT-IR analysis because they are not associated with vibrational modes in the infrared. Estimated biases due to undetected carbon fraction for these simulations are used to make adjustments in these carbon-centric metrics such that model-measurement differences are framed in terms of unmeasured heteroatoms (e.g., in hydroperoxide and nitrate groups for the case studied in this demonstration). The formality of this method provides framework for extending FG analysis to not only model-measurement but also instrument intercomparisons in other chemical systems.

## 1 Introduction

Organic aerosols are complex mixtures of thousands of different types of compounds that vary in structure and physicochemical properties. This diversity poses challenges for comprehensive characterization, even while estimates of overall mass abundance and its contributing factors are still desirable. Functional group (FG) analysis is an approach that presents a level of charac-





terization that provides a bridge between full molecular speciation, which is useful for precisely tracking specific classes of physical and chemical transformations, and elemental composition, which is useful for mass closure analysis. FGs are structural units in molecules that describe important condensed-phase interactions that contribute to properties like volatility and hygroscopicity, and FG analysis provides information useful for overall organic mass quantification and its apportionment by
source class in past studies (e.g., Russell et al., 2011). FGs are also central to understanding reactivity and resulting chemical transformations, and their characterization by measurement and in model simulation can provide a method of evaluating our understanding of functionalization (i.e., through bonding with heteroatoms) in organic aerosol mixtures. However, studies on this topic have thus far been very limited on account of challenges in quantitative characterization of FGs, which requires either advanced algorithms for spectral interpretation or derivitization steps for chemical analysis. In anticipation of continued
progress in analytical technology, Ruggeri and Takahama (2016) and Ruggeri et al. (2016) introduced a method for harvesting FG information from molecularly speciated measurements (e.g., gas chromatography-mass spectrometry, GC-MS; Rogge et al., 1993) and chemically explicit model simulation (e.g., Master Chemical Mechanism, MCMv3.2; Jenkin et al., 1997, 2003; Saunders et al., 2003).

In this study, we build upon the work by Ruggeri et al. (2016) to further improve our capability for model-measurement
intercomparison using FG analysis.Ruggeri et al. (2016) compared changes in relative molar abundances of FGs in chamber experiments measured by Fourier Transform Infrared (FT-IR) spectroscopy against composition simulated with a chemically explicit gas-phase reaction mechanism coupled to a gas/particle (G/P) partitioning module. As molar FG composition is directly obtained from measured FT-IR absorbances, this is a sensible metric used to track changes in chemical composition and has been used in other studies (e.g., Camredon et al., 2007). However, estimating FG contributions to carbon-centric metrics more
commonly used to characterize organic aerosol oxidation or mass yields, such as organic carbon (OC) and organic matter (OM) mass, OM/OC mass ratios, atomic ratios, and mean carbon oxidation state (Russell, 2003; Aiken et al., 2008; Kroll et al., 2011, 2015) is not straightforward. Central to this task is understanding which fraction of carbon atoms are "detected" by measurement of any given set of FGs, and estimating the overall carbon abundance from FGs without multiply counting the polyfunctional carbon atoms.

Some of these metrics have been calculated from FT-IR measurements by previous researchers based on assumptions regarding the underlying molecular structure (e.g., Allen et al., 1994; Maria et al., 2003; Reff et al., 2007; Russell et al., 2009; Chhabra et al., 2011). For instance, Chhabra et al. (2011) assumed bonding configurations in secondary organic aerosol (SOA) products to be consistent to the parent volatile organic compound (VOC) to estimate the carbon content from measured FG abundance. Ranney and Ziemann (2016) also use the number of carbon atoms in the parent VOC to normalize FG concentrations reported
for SOA mixtures. Russell (2003) introduced a functional group index (FGI) to conceptualize how OM/OC ratios varies according to chain length and functionalization for specific sets of compound classes, and provided an evaluation from mass spectrometry measurements that comprised up to 10% of the total OM mass. Using results from numerical simulation of SOA formation, we now describe methods for estimating carbon content based on molecular parameters that describe the underlying mixture composition consisting of a diverse set of polyfunctional compounds, and a means of examining dependence of



carbon-centric on composition without invoking knowledge about molecular chain lengths, which is not well characterized by FG analysis. The benefit of developing a systematic approach is that we can precisely understand the achievable mass recovery, and biases incurred on the calculated O/C and OM/OC for a given set of molecules and FGs analyzed (when extraction efficiencies are not invoked, OM mass recovery is primarily dependent on the completeness of FG calibration models constructed).

These estimates may then be used to propose mixture-specific adjustments to facilitate more direct intercomparisons with other data. This work will focus on FG abundances obtained by FT-IR measurements, but many aspects are generalizable to other types of FG analysis (e.g., Dron et al., 2010; Ranney and Ziemann, 2016).

The objective described above is addressed in this work by 1) conceptualizing SOA as a collection of carbon atoms that are functionalized in different ways, and 2) the FT-IR as a tool that measures some subset of such functionalized carbon

structures. These "carbon types" can be used to calculate the OM properties described above, and gives rise to observed FGs in measurement. Carbon type representation of complex mixtures has a strong precedent in the study of organic chemistry in the atmosphere. For example, the Carbon Bond Mechanism (Whitten et al., 1980) defines chemical reaction schemes according to reactivity of carbon atoms classified according to functionality, without regard to membership in a molecule. The "carbon vector" in GECKO-A (Aumont et al., 2005) is a description of functionalized carbon types and retains information regarding

transformations in functionalization (while a separate connectivity matrix tracks transformation in the carbon skeleton upon accretion or fragmentation). In the commonly used volatility basis set (VBS), changes in carbon mass are conserved according to functionalization by oxygen, nitrogen, or overall carbon oxidation state (Kroll et al., 2011, 2015; Donahue et al., 2012; Chuang and Donahue, 2016). Quantitative analysis of additional "groups" that describe the underlying skeletal (e.g., ring, aromatic, or unsaturated) structures that change with fragmentation and accretion reactions (Kroll et al., 2011) have not been

sufficiently advanced by FG analysis to provide complete estimates of mean molecular size and other aerosol properties that govern volatility and solubility (Zuend et al., 2008). However, past precedents mentioned above indicate that classification of carbon atoms according to extent of functionalization may have merit in harmonizing observations with model representations for calculating common mixture characteristics of OM.

In this work, we illustrate how measured FGs can be related to properties of various carbon types comprising a diverse set

of polyfunctional molecules. We use the proposed relationships to determine which carbon types are measured according to FGs included in calibration models, and biases resulting from partial analysis of the different carbon types in the mixture. For illustration, $\alpha$-pinene gas-phase photooxidation simulation in the presence of $NO_x$ with G/P partitioning is analyzed and compared against chamber experiments upon which the simulations were based. We will assume a perfect calibration where we assume flawless knowledge of the bond abundance to isolate biases due to measured and unmeasured carbon types.

Such a scenario is obviously not physically achievable, but serves as a convenient reference by which we can proceed with a meaningful model-measurement comparison.





## 2 Methods

After describing our data set in Section 2.1, we introduce a few relationships among FG, atomic composition, and carbon types in Section 2.2. We then describe how we can estimate whether a particular carbon type is detected by FT-IR based on the set of FG calibrations used and properties that we calculate as a result in Section 2.3. We then present methods for actually estimating

the number of polyfunctional carbon atoms from FG abundance to minimize multiple counting in Section 2.4. The code and software used in this and previous manuscripts are made available under the GNU Public License (Appendix A).

### 2.1 Data set

We focus this analysis on a specific a simulation scenario of Ruggeri et al. (2016) in which comparison of model results to reference measurements had the smallest discrepancy according to relative molar abundance of FGs, until model-measurement

agreement diverged on what was attributed to the role of heterogeneous chemistry and aging not implemented in the model. To briefly describe the simulation, the MCMv3.2 gas-phase chemistry module generated by the Kinetic Pre-Processor (Sandu and Sander, 2006; Henderson, 2015) was coupled with a gas/particle organic absorptive partitioning scheme via operator splitting (Yanenko, 1971). The SIMPOL.1 group contribution model (Pankow and Asher, 2008) was used to estimate the equilibrium vapor pressure for individual molecules, and the dynamics of mass transfer to a monodisperse particle population

were simulated using LSODE (Livermore Solver for Ordinary Differential Equations; Radhakrishnan and Hindmarsh, 1993). Wall losses of particles and semivolatile volatile organic compounds (SVOCs) were neglected. The scenario we further analyze for this study was defined by initial $\alpha$-pinene and $NO_x$ concentrations of 300 and 240 ppb, respectively. The relative humidity was fixed at 61%, which influenced the rate of $HO_2$ radical self reaction to form hydrogen peroxide, but water uptake and influence on G/P partitioning was not considered. The light intensity was fixed (Saunders et al., 2003) to be consistent with

experimental conditions. This scenario was labeled the "APIN-lNOx" simulation. In this work, we will refer to this as the APIN simulation, as we discuss none of the other scenarios and thus eliminate the need for an additional modifier to the label. To focus on a particular mixture, we select a reference period as the apex in SOA concentration occurring at 9.3 hours (labeled as $t_{\max SOA}$ ) of the 22 hour simulation as used by Ruggeri et al. (2016) to examine molecular contributions to overall SOA mass and FG abundance. With detailed knowledge of molecular structure and composition in this simulation, we apply the analysis

described in Sections 2.2–2.4.

The conditions for the simulations described above were selected to mimic chamber experiments in which FG composition was measured by Sax et al. (2005). Sax et al. (2005) collected particles between 86 and 343 nm onto (infrared-transparent) zinc selenide crystals by impaction, and samples were analyzed immediately afterward to minimize storage artifacts. Samples were scanned rapidly to minimize evaporative losses in the FT-IR sample compartment. Sax et al. (2005) report that repeated

analysis of the same samples by FT-IR yielded consistent results, suggesting robustness in reported values. Samples collected during 3.1–4.2 hours and 17.6–21.6 hours (which we label as "4h" and "21h", respectively) were selected by Ruggeri et al. (2016) for comparison against model simulation for the corresponding periods, and we will follow this convention here.



Only relative metrics are used as Sax et al. (2005) reported measurements in mole fractions of FGs, and the simulations do not include wall losses of particles and SVOCs that affect overall estimates of yield. Neglecting compound-specific SVOC deposition to walls may further incur biases in relative compositions as raised by Ruggeri et al. (2016), but for this conceptual study we neglect its effect as its parameters are not precisely known.

## 2.2 Definitions

Given molar concentration of molecules $\boldsymbol{n}_{\mathrm{molec}} = [n_i]$ in a mixture (consisting of $\mathcal{M}$ molecules) and group composition matrix $\mathbf{X} = [x_{ij}]$, FT-IR analysis measures the total abundance of bonds $\boldsymbol{n}_{\mathrm{group}} = [n_j]$ for each FG in $\mathcal{J}$. We write this in scalar notation as

$$n_j = \sum_{i \in \mathcal{M}} n_i x_{ij} \quad \forall j \in \mathcal{J} . \tag{1}$$

$n_j$ is the observed quantity from measurement, and represents the sum of functional group composition of molecules weighted weighted by their molar abundance.

A statement of atom balance is enabled by the group-atom matrix $\boldsymbol{\Lambda} = [\lambda_{aj}]$ (Takahama et al., 2013) by relating $n_j$ to the atomic abundance $\boldsymbol{n}_{\mathrm{atom}} = [n_a]$ in the mixture:

$$n_a = \sum_{j \in \mathcal{J}} \lambda_{aj} n_j , \tag{2}$$

However, the fact that the same polyfunctional carbon atom can be associated with several FGs poses challenges for reasoning out $\lambda_{\mathrm{C},j}$ for carbon. Therefore, we introduce a carbon type matrix $\mathbf{Y} = [y_{ik}]$ that enumerates the composition of each molecule in terms of specific number of carbon types, and a carbon-group matrix $\boldsymbol{\Theta} = [\theta_{kj}]$ that relates each carbon type to its unique structure of functionalization.

A statement of FG balance can be constructed from the carbon type matrix, carbon-group matrix, and group composition matrix:

$$\sum_{k \in \mathcal{C}} y_{ik} \theta_{kj} = x_{ij} \quad \forall i \in \mathcal{M}, j \in \mathcal{J} . \tag{3}$$

Conversely, a statement of carbon type balance can be made by introducing a matrix, $\boldsymbol{\Phi} = [\phi_{jk}]$ from which carbon type abundance can be obtained with FG abundance to construct a statement of carbon type balance:

$$y_{ik} = \sum_{j \in \mathcal{J}} x_{ij} \phi_{jk} \quad \forall i \in \mathcal{M}, j \in \mathcal{J} . \tag{4}$$

A minimal illustration for two simple molecules, ethane and ethanol, are shown in Fig. 1. Symbols are tabulated in Table B1. Explanation of additional arrays $\boldsymbol{\Lambda}$ (atom-group matrix), $\boldsymbol{\zeta}$ (carbon oxidation state vector), and $\boldsymbol{z}$ (oxidation state contribution





vector) completing the atom and oxidation state balance follow below. In contrast to concise expressions expressed in Figure 1, we continue with use of scalar notation blow to more conveniently invoke element-wise, row-wise, and column-wise summations, but will return to array notation for describing solutions to system of equations (Section 2.4).

In our APIN mechanism, there are 327 molecules, 22 FGs, and 41 carbon types (Figure 2), though several are associated with
radical structures or unusual structures that are not found in the most abundant compounds. These do not contribute to the organic aerosol mass, but is included for a complete description of the APIN mechanism. Furthermore, while the equalities introduced in Figure 1 are formulated to hold at the level of individual molecules, we demonstrate their application in describing the underlying relationships in molecular mixtures.

The carbon type matrix provides a conceptual relationship for relating FGs to number of carbon atoms in a mixture (equation
2 for carbon is also restated on the right hand side):

$$n_C = \sum_{i \in \mathcal{M}} \sum_{k \in \mathcal{C}} n_i y_{ik} = \sum_{i \in \mathcal{M}} \sum_{j \in \mathcal{J}} n_i \lambda_{C,j} x_{ij} \tag{5}$$

and we can see from equations 4 and 5 that $\lambda_{C,j}$ is equivalent to the column-wise summation of $\phi_{jk}$.

$$\lambda_{C,j} = \sum_{k \in \mathcal{C}} \phi_{jk} \quad \forall\, j \in \mathcal{J}. \tag{6}$$

Previous values for $\lambda_C$ are shown in Table 1. The atomic abundance for each carbon type $k$ is calculated as $n_{ka} = \sum_{j \in \mathcal{J}} \lambda_{aj} \theta_{kj}$,
as follows from equation 3 and 2.

The mean carbon oxidation state can be estimated from: 1) $y_{ik}$ through the oxidation state $\boldsymbol{\zeta} = [\zeta_k]$ specific to carbon type, and 2) $x_{ij}$ and individual FG contributions $\boldsymbol{z} = [z_j]$ to carbon oxidation state:

$$\overline{OS}_C = \frac{1}{n_C} \sum_{i \in \mathcal{M}} \sum_{k \in \mathcal{C}} n_i y_{ik} \zeta_k = \frac{1}{n_C} \sum_{i \in \mathcal{M}} \sum_{j \in \mathcal{J}} n_i x_{ij} z_j \tag{7}$$

From equation 3, we can see that $\zeta_k$ and $z_j$ are related through the following equality:

$$\zeta_k = \sum_{j \in \mathcal{J}} \theta_{kj} z_j \quad \forall\, k \in \mathcal{C} \tag{8}$$

All elements in equation 3 can be known precisely for any set of molecules $\mathcal{M}$ from the chemometric patterns and atom-level validation described by Ruggeri and Takahama (2016), and is summarized in Section S1. Solution methods for $\phi_{jk}$ and $\lambda_{C,j}$ are presented in 2.4.

## 2.3   Theoretical mass recovery and estimated properties

This section describes methods for determining if the carbon type is detected by FT-IR and how relationships introduced in Section 2.2 can be modified for a more direct comparison with measurements. The main idea is to consider only the subset of





carbon atoms which is bonded to any of the FGs measured in a given experiment, and analyze properties only for those carbon atoms as to what is the achievable degree of characterization of the SOA.

Given a set of FG which are measured $\mathcal{J}^* \subseteq \mathcal{J}$ and the corresponding subset of carbon atoms $\mathcal{C}^* \subseteq \mathcal{C}$ which only contain these FGs, we can estimate the number of carbon measured from a modification of equation 5:

$$5 \quad n_{\mathrm{C}}^* = \sum_{i \in \mathcal{M}} \sum_{k \in \mathcal{C}^*} n_i y_{ik} = \sum_{i \in \mathcal{M}} \sum_{k \in \mathcal{C}} n_i y_{ik} \cdot \mathrm{sgn} \left( \sum_{j \in \mathcal{J}^*} \theta_{kj} \right) . \tag{9}$$

sgn is the signum function, which will return 0 when its argument is 0 (no FGs associated with carbon type $k$ are in the measured set) and 1 when its argument is positive (one or more FGs belong to the measured set). The total carbon recovery is calculated as $n_{\mathrm{C}}^*/n_{\mathrm{C}}$.

We consider three sets of FGs for $\mathcal{J}^*$. Set1 = {aCH, aCOH, COOH, ketone and aldehyde carbonyl, $\mathrm{CONO_2}$}, and comprises
10 FGs reported by Sax et al. (2005) and many others (e.g., Maria et al., 2003; Coury and Dillner, 2008; Russell et al., 2009; Day et al., 2010). Set2 = Set1 + {eCH hydroperoxide, peroxyacyl nitrate}, and comprises Set1 and three additional FGs that are not commonly reported for OM characterization but have medium to strong absorption bands in the mid-infrared wavelengths (Appendix C) (not inclusive) and relevant for this system. The set labeled as Full comprises all groups present in OM, including quaternary and tertiary sp$^2$ carbon (carbon atoms that are only bonded to other carbon atoms) that accounts for 7% of the mass
in the APIN simulation at $t_{\mathrm{max\,SOA}}$ , and also the remaining groups (Figure 2) that accounts for <1% of the remaining mass.

We can estimate OM as the sum of elements multiplied by their respective molecular weights using equation 2. Atomic ratios are calculated as $n_a/n_{\mathrm{C}}$ for all heteroatoms $a = \{\mathrm{H,N,O}\}$ (S is not included in this chemical mechanism, but this principle can be extended for mechanisms that include it):

$$n_a^* = \sum_{j \in \mathcal{J}*} \lambda_{aj} n_j . \tag{10}$$

Atomic ratios are calculated as $n_a^*/n_{\mathrm{C}}^*$

To estimate the mean carbon oxidation state, we can replace $n_{\mathrm{C}}$ with $n_{\mathrm{C}}^*$ and sum over $\mathcal{J}^*$ instead of $\mathcal{J}$ in equation 7 by corollary with equation 9:

$$\overline{\mathrm{OS}}_{\mathrm{C}} \approx \frac{1}{n_{\mathrm{C}}^*} \sum_{j \in \mathcal{J}^*} z_j n_j . \tag{11}$$

## 2.4 Estimation of carbon abundance

In this section, we describe methods for estimating $n_{\mathrm{C}}$ from measured abundance of FGs. The main objective is to arrive at a set of coefficients $\hat{\lambda}_{\mathrm{C}}$ that, when multiplied by FG abundance $n_j$ for measured FGs $\mathcal{J}^*$, provides an estimate $\hat{n}_{\mathrm{C}}^*$ that does not count multiples of the same carbon atoms which are attached to the suite of FGs analyzed:

$$\hat{n}_{\mathrm{C}}^* = \sum_{j \in \mathcal{J}^*} \hat{\lambda}_{\mathrm{C},j} n_j . \tag{12}$$





The use of the hat over a symbol denotes a statistically estimated quantity.

It is convenient to continue discussion of solutions to a system of equations in array notation (similar to what is used in Figure 1). Let $\mathbf{Y} = [n_i y_{ik}]$, $\mathbf{X} = [n_i x_{ij}]$, $\boldsymbol{\Theta} = [\theta_{kj}]$, $\boldsymbol{\Phi} = [\phi_{jk}]$, $\lambda_{\mathrm{C}} = [\sum_{k \in \mathcal{C}} \phi_{jk}]$, and $\boldsymbol{n}_{\mathrm{C}} = [\sum_{k \in \mathcal{C}} y_{ik}]$. The FGs and carbon type abundances can be written as $\mathbf{Y}\boldsymbol{\Theta} = \mathbf{X}$. The most obvious solution is to take the generalized or Moore-Penrose inverse, $\hat{\boldsymbol{\Phi}} = \boldsymbol{\Theta}^{+}$. In the example illustrated in Figure 1, the solution to $\boldsymbol{\Phi} = \boldsymbol{\Theta}^{-1}$ and $\boldsymbol{\lambda}_{\mathrm{C}}$ (a row of $\boldsymbol{\Lambda}^{T}$) using such an approach is provided. While exact solutions can be found for this illustration because $\boldsymbol{\Theta}$ is square (the number of carbon types equals the number of types of FGs), the pseudo-inverse solution will not be meaningful in a more general case as the number of ways in which FGs are arranged on carbon atoms exceeds the number of measured FG used for discrimination. Therefore, while carbon types are a useful concept to describe the underlying representation of functionalized organic compounds, it is generally not possible to retrieve the exact abundance of each carbon type from FG measurements. To arrive at an approximate solution for estimation of the total carbon atoms without discrimination of carbon types, we consider the three approaches described below.

First, we consider each carbon type in isolation ("COUNT" method) and average the reciprocal of measured FGs per carbon enumerated for each carbon type.

$$\hat{\lambda}_{\mathrm{C},j} = \frac{1}{|\mathcal{C}_j|} \sum_{k \in \mathcal{C}_j} \frac{1}{\sum_{j' \in \mathcal{J}^*} \theta_{kj'}} \tag{13}$$

$|\cdot|$ denotes the cardinality of (i.e., number of elements in) a set and $\mathcal{C}_j$ is the set of carbon types in which FG $j$ appears, and is the origin of the dependence of $\lambda_C$ on $j$. The rationale can be supported by the illustration (Figure 1) in which 1/3 for $\boldsymbol{\lambda}_{\mathrm{C}}$ reflects the number of measured FGs attached to each carbon atom.

In the second approach ("COMPOUND" method), we find $\boldsymbol{\Phi}$ that corresponds to the least squares solution to the following equation:

$$\hat{\mathbf{Y}} = \mathbf{X}\hat{\boldsymbol{\Phi}} . \tag{14}$$

$\hat{\boldsymbol{\lambda}}_{\mathrm{C}}$ is found by row-wise summation of $\hat{\boldsymbol{\Phi}}$ (equation 6) (which is also equivalent to solving for $\hat{\boldsymbol{\lambda}}_{\mathrm{C}}$ directly in the reduced expression, $\hat{\boldsymbol{n}}_{\mathrm{C}} = \boldsymbol{X}\hat{\boldsymbol{\lambda}}_{\mathrm{C}}$). Given the wide range of possibilities in composition, we set molar abundances to unity such that each compound within each group (SVOC) is uniformly weighted. We average over carbon types present in molecules relevant to certain mixture classes with uniform weighting such that the derived coefficients are not overly specific to any particular mixture.

In the third approach ("MIXTURE" method), we reformulate $\mathbf{Y} = [n_{mi} y_{ij}]$ and $\mathbf{X} = [n_{mi} x_{ij}]$ such that its rows contain the FG abundance of the mixture of each time step $t_m$ of the a-pinene photooxidation simulation, and $\boldsymbol{\lambda}_{\mathrm{C}}$ is found by fitting $\mathbf{X}$ to $\boldsymbol{n}_{\mathrm{C}}^*$, the time series of carbon atom concentration at each time step. For MIXTURE, we use a constrained least squares approach where the values of the regression coefficients are bounded between 0 and 1 as the coefficients for FGs with low abundance (e.g., eCH and CONO2) are not well constrained (the solution is insensitive to their values).



Numerical details aside, the main differences among the three are the data sets used for estimation. COUNT uses information from $\Theta$ only (defined for the FGs in the APIN mechanism), COMPOUND uses carbon type abundances in compounds (limited to SVOCs in the APIN mechanism), and MIXTURE uses mixture information (from different periods in the APIN simulation). Each of the solutions produces a series of irrational numbers that may be overly precise for the data set used for estimation.

As later shown, we will also adjust the COUNT solutions to rational values of {1/4, 1/3, 1/2, 1} (with exception for $\lambda_{C,aCH}$ which we fix to a value of 0.45 as explained in Section 3.3), and we will refer to this as the "NOMINAL" solution. For the COMPOUND and MIXTURE methods, FGs and carbon types with a unique (one-to-one) correspondence (e.g., carbon atoms associated with carboxylic acid and ketonic and aldehydic carbonyl groups) are excluded from the fitting, as their coefficients are known unambiguously. Evaluations of estimates are expressed as a ratio of the estimate over the reference

value: $\hat{n}_C^*/n_C^*$. We remark that we focus on harvesting information from the APIN simulation results only, but these methods can (and should) be applied to study abundances in molecular speciation data from chamber experiments under different oxidation and environmental conditions (e.g., Yu et al., 1999; Glasius et al., 2000) in future work.

## 3 Results

We first describe the APIN simulation results of Ruggeri et al. (2016) recast in terms of abundance of carbon types in Section

3.1. We then describe mass recovery and biases in property estimates due solely to unmeasured carbon atoms in Section 3.2. In Section 3.3, we describe results from applying different methods for estimating carbon abundance from measured FGs. Finally, in Section 3.4, we present estimates of properties from FG measurements and compare to model simulations.

### 3.1 Evolution of carbon types

The time series of carbon type abundance is shown by its contribution fraction for each time period in Figure 3, and the carbon

type composition of the most abundant molecules at $t_{\max SOA}$ is depicted in Figure 4. Descriptions for the carbon types found in $t_{\max SOA}$ are shown in Figure 2. We observe that changes in carbon type composition is rapid within the first four hours, but generally changes much more slowly after this period. Many of the dominant carbon types are generally similar between the gas and aerosol phases and include: methyl ($CH_3$), methylene ($CH_2$), ketone, primary alcohol, and secondary alcohols, acid (COOH), hydroperoxides, and peroxyacyl nitrate groups. However, the order of abundance is different between phases — for

instance, the peroxyacyl nitrate is more abundant in the gas phase (carbon type 10; Figure 2). As visualized in Figure 4 and described by Ruggeri et al. (2016), the molecular abundance is dominated by a small number of polyfunctional compounds (out of the [200] compounds in the mechanism), so their carbon types are weighted heavily in the overall carbon type composition.





## 3.2 Theoretical mass recovery and property estimation

The ordered contribution to mass recoveries of OC and OM for the most dominant carbon types at $t_{\mathrm{max\,SOA}}$ are displayed in Figure 5. Greater than 99.9% of the OC and OM mass is accounted for by 15 carbon types during this period, while more than 20 compounds are required to reconstruct aerosol OC mass with >99.9% recovery (Figure 4)). Mass recovery with Set1

is on the order of 80%. The fraction of OC estimated by FT-IR relative to OC measured by thermal optical methods are often within a similar range (e.g., Maria et al., 2003; Ruthenburg et al., 2014). With additional bonds in Set2, 93% carbon recovery is achieved. The unmeasured carbon types are quaternary and tertiary sp$^2$ carbon that are bonded to C-bonds only, and together comprise 7% of the OC (Full case).

Going from Set1 to Set2, the increase in fraction of recovered OM is greater than recovered OC because of the hydroperoxide

and peroxyacyl nitrate mass is much greater than the mass of carbon bearing these FGs. The resulting effect on estimated properties is shown in Figure 6. H/C recovery is high for Set1 already, but we are missing the oxygen from hydroperoxide and peroxyacyl nitrate. eCH is small. N/C is very small (low $NO_x$ conditions). OM/OC can be off by 0.2. Even with nearly full mass recovery, ratios are often inflated by a small amount on account of the unmeasured carbon (i.e., $n_C^* \leq n_C$).

The carbon oxidation state distribution and recoverable portions for $t_{\mathrm{max\,SOA}}$ are shown in Figure 7a. This figure visually

reinforces the abundance of methyl carbons ($CH_3$, $OS_C$ = -3), methylene carbons ($CH_2$, $OS_C$ = -2) discussed above, though there are other carbon types contributing to the $OS_C$ = -2 category (Figure 2). The unmeasurable carbon types with FT-IR are those with $OS_C$ = 0, which are the quaternary and tertiary sp2 carbon (carbon types which are measurable in the $OS_C$ = 0 category have a balance of negative and positive values from aCH and electronegative heteroatoms). The value of the additional FGs in Set2 are for characterization of oxidizing FGs (hydroperoxide and peroxyacyl nitrate) that on carbon atoms

with overall oxidation states of 1 and 3. Estimates of the mean $\overline{OS}_C$ is shown in Figure 7, panel (b). We can see that the bias in estimation for neglecting hydroperoxide and peroxyacyl nitrate is not as great as for the O/C ratio, since the $OS_C$ is determined by the atom and bond connected to the carbon atom directly, and the rest of the multiple oxygen atoms in the FG are not considered. The 2O/C-H/C estimate commonly used with elemental analysis will lead to a slight overestimation of the $\overline{OS}_C$ in the event that oxygen single-bonded to carbon (hydroxyl and hydroperoxide groups) exist in large abundance proportionally to

double-bonded carbonyl groups (Kroll et al., 2011).

## 3.3 Estimation of carbon abundance

Table 2 summarizes the new values for $\hat{\lambda}_C$ obtained by the different estimation methods described in Section 2.4. Comparison of $\hat{n}_C^*$ estimated using these values against $n_C^*$ in individual compounds is shown in Figure 8, and the comparison of $\hat{n}_C^*$ and $n_C^*$ in overall aerosol mixtures at different time periods in the APIN simulation is shown in Figure 9.

Values for $\hat{\lambda}_C$ are roughly similar among estimation methods, with exception to the MIXTURE estimate. Overall, we find that the coefficient for aCH is close to but less than the often assumed value of 0.5 (Table 1), which can play an important role on





account of the abundance of aCH bonds and carbon types associated with aCH. For the MIXTURE estimate, $\hat{\lambda}_{C,aCH} = 0.5$ but is balanced by exceptionally small coefficients for aCOH and hydroperoxide. This combination of coefficients essentially downweights the contributions from carbon types associated with aCH and hydroperoxide, which we know to be present in abundance (within top 6 for the APIN simulation at $t_{\max SOA}$, but remains significant throughout the simulation as seen in Figure 3). Therefore, we conclude that the estimates obtained for this fit are statistically convenient but less physically relevant than the other estimates. For the NOMINAL case, we fix the aCH to $\lambda_{C,aCH} = 0.45$ and the rest to the nearest rational numbers.

For individual compounds, we note that using either Set1 and Set2 reproduce $n_C^*$ with similar biases on average: 11% for COUNT and within 4% for the others. COUNT underestimates $n_C^*$ in large compounds with lower oxidation states containing many aCH groups, because of the low estimate of $\hat{\lambda}_{C,aCH}$. COMPOUND reproduces $n_C^*$ well because this is the data set COMPOUND was fit to, but MIXTURE also does well. The NOMINAL solution also does well, but largely owing to the $\lambda_{C,aCH}$ adjustment.

For reproducing mixture composition, trends in biases are similar to individual compounds, with underestimation by as much as 18% for COUNT and within 7% for the other estimation methods. MIXTURE performs the best because this is the data set it was fitted to, but we see that the COMPOUND and NOMINAL are also acceptable. There is generally a trend toward increasing $\hat{n}_C^*/n_C^*$ over the duration of the simulation, which indicates an evolving relationship between FGs and carbon abundance with mixture composition. Time-dependent (i.e., mixture-specific) estimates of $\lambda_C$ may be warranted when the change in composition becomes more significant.

We therefore conclude that errors for estimation of $n_C^*$ can be quite low and are well below 10% according to our evaluation. Even a 10% error in estimation of $n_C^*$ will lead to a 9% error in the estimation of any individual atomic ratio, and 5% estimation in the OM/OC ratio (Appendix D). Therefore, in applying the NOMINAL coefficients to measured values of FGs under conditions upon which the APIN simulations were based (Section 3.4), we discuss deterministic explanations for model-measurement discrepances with less consideration toward statistical estimation error of $n_C^*$.

## 3.4 Comparison with measurements

In this section, we discuss O/C, OM/OC, and $\overline{OS}_C$ estimated from measurements ending at hours 4 and 21 and APIN simulation results integrated over the same periods (Figure 10). We label the interpretation of measurements with previous estimates of $\lambda_C$ (Table 1) as "MEAS-PREV", measurements with revised estimates of $\lambda_C$ (Table 2) as "MEAS-NOM", simulation results using FGs from Set1 as "SIM-SET1", and full simulation results as "SIM-FULL"; further adjustments are made for the last three estimates as justified next. In Section 3.2, we presented an estimate of mass recovery ($n_C^*/n_C$) and how this led to biased estimates of atomic ratios and OM/OC ratio. In Section 3.3, we also showed that we can derive estimates of $\lambda_C$ such that errors in estimation of $n_C^*$ was small (i.e., $\hat{n}_C^*/n_C^*$ near unity). Therefore, for the following comparisons, we neglect the latter error and correct biases due to carbon mass recovery by using our best estimate of $n_C$, rather than $n_C^*$, as the normalization factor. The proportion of detected carbon to make this correction is obtained from SIM-SET1 in which the same FGs as





measurements are used. While the adjustment is only approximate on account of differences in the real experimental system and model simulation, it reduces systematic biases in carbon-centric metrics as described in Section 3.2 such that deviations from true ratios can be largely attributed to the unmeasured heteroatoms. For MEAS-NOM, the atomic ratio is then estimated as $n_{\mathrm{a}}^*/n_{\mathrm{C}} = n_{\mathrm{a}}^*/n_{\mathrm{C}}^* \times (n_{\mathrm{C}}^*/n_{\mathrm{C}})_{\text{SIM-SET1}}$ and the OM/OC and $\overline{\mathrm{OS}}_{\mathrm{C}}$ by similar adjustment. MEAS-PREV remains unadjusted to

be used as a reference estimated without prior knowledge about the underlying molecular structures of the SOA products.

First, we remark on differences for estimated metrics from two sets of coefficients applied to the same FG measurements. MEAS-PREV overestimates the $n_{\mathrm{C}}^*$ compared to MEAS-NOM by 21-28% on account of higher $\lambda_{\mathrm{C}}$ coefficients used in the former. However, the uncorrected bias due to lower mass recovery of carbon is approximately the same magnitude, and ultimately leads to ratioed values (O/C, H/C, OM/OC, $\overline{\mathrm{OS}}_{\mathrm{C}}$ ) similar to MEAS-NOM. While it is not clear that $\lambda_{\mathrm{C}}$ derived in this

work accurately represents the true mixture, we posit that the degree of functionalization characterized by the new estimate is likely to be more representative for the product mixture after successive oxidation of the APIN, rather than APIN itself (as assumed by MEAS-PREV). Chhabra et al. (2011) report O/C and H/C estimates from FT-IR using coefficients of MEAS-PREV and found that they were within range of AMS values; this is possibly due to the offsetting of errors as demonstrated here. In further discussion, we will discuss the interpretation of observations based on MEAS-NOM.

MEAS-NOM and SIM-SET1 are the two estimates intended to provide the most direct comparison between experiment and numerical simulation. While the discrepancy in carbonyl and carboxyl groups at 4 hours is only 2% and 3% in mole fraction, respectively (Ruggeri et al., 2016), this leads to an overall discrepancy of 0.16 for O/C and 0.2 for OM/OC. Since aCOH, carbonyl, and COOH groups are a larger contributor to the mass relative to the aCH group, discrepancies in molar abundance of oxygenated FGs are magnified when represented in OM/OC ratios and can have a non-negligible influence on interpretation

of mass yields. After 21 hours, the difference is 0.38 in O/C and 0.48 in OM/OC. Ruggeri et al. (2016) attributed the apparent divergence to mechanisms not included in the model. Oligomerization was not considered a likely candidate as this process not expected to contribute to increased oxygenation reported by FT-IR. Condensed-phase photolysis can lead to conversion of hydroperoxides to carbonyls (some of which are lost to the vapor phase as more volatile molecules) (Epstein et al., 2014), but even a hypothetical full molar conversion is insufficient to explain the model-measurement differences in carbonyl groups

(Ruggeri et al., 2016). Other missing mechanisms may include autoxidation (Crounse et al., 2013) which can produce extremely low volatility (ELVOC; Ehn et al., 2014) or highly oxygenated molecules (HOM; Tröstl et al., 2016) in the gas phase, or radical reactions in the condensed phase that lead to highly oxidized products (Lim et al., 2010) containing these measured FGs. In these comparisons, we cannot rule out that some biases in measurement may originate from molar absorption coefficients estimated for each FG in FT-IR. The absorption intensity is determined by a change in the magnitude of the dipole moment

and can vary according to molecule or mixture environment; the representativeness of applied absorption coefficients in these SOA mixtures is a possible area for future inquiry. However, Takahama et al. (2013) cite variations on the order of 20% for oxygenated FGs in several carboxylic acid, and ketone species, which provide some constraints on this uncertainty for the range of compound classes evaluated in their study.



As reported by Ruggeri et al. (2016), SIM-FULL has similar O/C of observations in similar chamber studies where Aerosol Mass Spectrometer (AMS) measurements were available (Chen et al., 2011; Zhang et al., 2015). OM in MEAS-NOM is less functionalized than in SIM-FULL at hour 4, but the opposite is true at hour 21 even while hydroperoxide and peroxyacyl nitrate is not included. The rate of transformation of these FGs remains uncertain, though Epstein et al. (2014) reports of lifetime of

hydroperoxide of approximately 6 days under summertime conditions in Los Angeles. Using the estimates of MEAS-NOM, the additional oxidation and aging process between 4 and 21 hours leads to an increase in O/C of about 0.24, including a 0.09 difference in O/C from carbonyl (a product of hydroperoxide photolysis). If we extrapolate the O/C of MEAS-NOM to that which includes hydroperoxide and peroxyacyl nitrate groups by assuming the same hydroperoxide and peroxyacyl nitrate contributions from SIM-FULL, we would obtain an overall O/C ratio of 0.7 at hour 4 and 0.9 at hour 21. The latter value is at the

higher end of O/C values by reported by AMS (e.g., Aiken et al., 2008; Jimenez et al., 2009; Lambe et al., 2015). A concurrent measurement of overall O/C and O/C partitioned by measured FG may provide better constraints on our understanding of OM transformations.

As with O/C and OM/OC, $\overline{OS}_C$ also highlights the greater extent of functionalization in observations than in simulations between hours 4 and 21. $\overline{OS}_C$ estimated from MEAS-NOM is in the range of low-volatility oxygenated organic aerosol (LV-OOA)

(Donahue et al., 2012), while they are in the range of semi-volatile oxygenated organic aerosol (SV-OOA) in the simulations as consistent with the species included in the MCMv3.2 mechanism. In simulation, the products found in the aerosol phase are contain more than six carbon atoms, and the smaller, highly oxidized molecules remain in the gas phase (Figure S1) As discussed in Section 3.2 and shown in comparison between SIM-MEAS1 and SIM-FULL (Figure 10c), the missing contributions from hydroperoxide and peroxyacyl to $\overline{OS}_C$ are likely to be small as only the valence of the bonded atoms, and not the total

atomic count of the FGs, contribute to the carbon oxidation state.

## 4   Conclusions

This study extends the work of Ruggeri and Takahama (2016) and Ruggeri et al. (2016) to demonstrate how molecular structure — specifically, functionalization — can inform comparisons between model and measurement through knowledge of the underlying carbon type abundances. For a measured subset of molar FG abundances, we estimate the expected mass recovery of

simulated OC and OM, and how this impacts reported properties such as atomic ratios (O/C, H/C) and OM/OC mass ratios that are of interest to the atmospheric aerosol community. Furthermore, we show how information regarding the underlying molecular structure can be used to better constrain the abundance of polyfunctional carbon that can be estimated from measurements of FGs.

For the $\alpha$-pinene photooxidation simulation analyzed, we find that 80% of the carbon is detectable by the set of commonly

measured FGs, and 7% is unmeasurable on account of having only carbon-carbon bonds. The problem of multiply enumerating polyfunctional carbon atoms using FG abundances for types in this simulated mixture introduces a smaller error, typically less than 10%. The coefficients required to map FG abundance to carbon abundance varies slightly from what has been assumed for





**Table A1.** Code.

| Name | Description | Repository |
| --- | --- | --- |
| *Substructure Search Program* | Enumerates FGs in molecules. | https://github.com/stakahama/aprl-ssp |
| *KPP with G/P Partitioning* | Generates model for gas phase chemistry with partitioning based on MCM mechanism. | https://github.com/stakahama/aprl-kpp-gp |
| *Carbontype analysis* | Maps to FGs to carbon types. Reproduces analysis and figures in this manuscript. | https://github.com/stakahama/aprl-carbontypes |

ambient samples; until more studies are conducted there may be reason to continue using previous coefficients for consistency. Comparison of simulation results to measured O/C, OM/OC, and carbon oxidation state partitioned by FG contributions elucidated the magnitude of missing LV-OOA (among other classes of molecules) in our model on these widely use metrics. Our current model only includes gas-phase chemistry prescribed by MCMv3.2 combined with gas-particle partitioning at present

time, but such comparisons can be extended as additional mechanisms are added. Within the context of this framework, the value of improving our knowledge of SOA formation and aging, investigating measurement artifacts, and developing calibration models for additional FGs for improved comparison with models can be better evaluated.

In that FG analysis measures characteristics of carbon types present in molecules of complex SOA mixtures, it can bridge our understanding of the atomic composition (e.g., measured via AMS) and constituent molecules identified by the growing

number of emerging analytical methods (e.g., Kalberer et al., 2006; Altieri et al., 2008; Jokinen et al., 2012; Chan et al., 2013; Chhabra et al., 2015; Lopez-Hilfiker et al., 2015; Nozière et al., 2015) to place their contributions in perspective. With regards to numerical simulation, model-measurement integration using FGs can further guide development of chemical mechanism generators (e.g., Aumont et al., 2005; Fooshee et al., 2012; Gao et al., 2016) and detailed benchmark models (e.g., Saunders et al., 2003), upon which reduced chemical reaction schemes are based (e.g., Dawson et al., 2016). We anticipate that the work

expounded in this series of manuscripts will strengthen the ensemble of tools available to study the complex phenomena of organic aerosol formation and aging.

## Appendix A: Code and software

Code and software associated with Ruggeri and Takahama (2016), Ruggeri et al. (2016), and this work are released under the GNU Public License (GPLv3) and listed in Table A1. The code can be downloaded as a zipped file from the listed repositories,

or via command line by the syntax `git clone https://github.com/stakahama/{reponame}`. Instructions are included in the README.md file in each repository. The corresponding author can be contacted for more information.

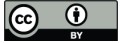

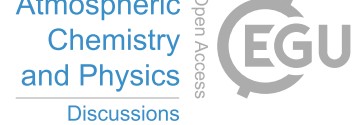

## Appendix B:  Notation

Symbols used throughout this manuscript are summarized in B1. Indices are written in lower case, vectors (single-column matrix) in bold italic, matrices in bold, and sets in calligraphy font. A hat over a variable indicates its statistically estimated value. A starred symbol indicates the detectable value corresponding to any given set of FGs.

**Table B1.** Mathematical symbols used in the manuscript and their descriptions.

| Category | Symbol | Description |
|---|---|---|
| Indices | $i$ | compound or molecule index |
| | $k$ | carbon type index |
| | $j$ | FG index |
| | $a$ | atom index |
| Variables | $n$ | number of moles of a substance (atom, compound, or FG) |
| | $\mathbf{X} = [x_{ij}]$ | group composition matrix |
| | $\mathbf{Y} = [y_{ik}]$ | carbon type matrix |
| | $\mathbf{\Theta} = [\theta_{kj}]$ | carbon-group matrix |
| | $\mathbf{\Phi} = [\phi_{jk}]$ | group-carbon matrix |
| | $\boldsymbol{\zeta} = [\zeta_k]$ | carbon type oxidation state vector |
| | $\boldsymbol{z} = [z_j]$ | oxidation state contribution vector |
| | $\mathbf{\Lambda} = [\lambda_{aj}]$ | atom-group matrix |
| | $\boldsymbol{\lambda}_{\mathrm{C}} = [\hat{\lambda}_{\mathrm{C},j}]$ | carbon atom-group vector |
| | $\mathrm{OS}_{\mathrm{C}}$ | carbon oxidation state |
| | $\overline{\mathrm{OS}}_{\mathrm{C}}$ | mean carbon oxidation state |
| Sets | $\mathcal{A}$ | set of atoms |
| | $\mathcal{M}$ | set of molecule types |
| | $\mathcal{J}$ | set of FGs |
| | $\mathcal{C}$ | set of carbon types |

## 5  Appendix C:  Vibrational modes

Absorption bands for additional FGs in Set2 (Section 2.3) are shown in Table C1. Hydroperoxide in the condensed phase has been measured using FT-IR (e.g., Shreve et al., 1951; van de Voort et al., 1994), but peroxyacyl nitrate analysis has mostly been limited to the gas phase (e.g., Gaffney et al., 1984; Monedero et al., 2008).





**Table C1.** Absorption bands in the mid-infrared for vibrational modes present in FGs proposed for Set2 (Section 2.3).

| FG | $\tilde{\nu}$ (cm$^{-1}$) | description |
|---|---|---|
| eCH[1] | 3005–2980 | C-H stretch |
| hydroperoxide[2] | 3300–3400 | OO-H stretch (strong) |
|  | 860–840 | O–OH stretch (weak) |
| peroxyacyl nitrate[2,3] | 760–849 | NO scissoring |
|  | 1340–1223 | NO$_2$ symmetric stretch |
|  | 1777–1700 | NO$_2$ anti-symmetric stretch |
|  | 1880–1777 | C=O stretch |

[1] Maria et al. (2003); [2] Shurvell (2006); [3] Monedero et al. (2008)

## Appendix D: Error estimation

In this section, relative uncertainties arising from the deviation between $\hat{n}_{\mathrm{C}}^*$ and $n_{\mathrm{C}}^*$ are translated into uncertainties of atomic ratios and OM/OC. As abundances of heteroatoms are determined from FG measurement do not suffer from multiple counting, uncertainties in their abundances are not considered.

5  Any of the estimation methods for $n_{\mathrm{C}}^*$ incurs a deviation from its true value by $\epsilon$, which we write as $\hat{n}_{\mathrm{C}}^* = n_{\mathrm{C}}^* + \epsilon$. We can recast this deviation as a relative error $\delta_{[n_{\mathrm{C}}^*]}$ with respect to $n_{\mathrm{C}}^*$ such that $\epsilon = \delta_{[n_{\mathrm{C}}^*]} n_{\mathrm{C}}^*$. The magnitude of $\delta_{[n_{\mathrm{C}}^*]}$ can be associated with the ratio $\hat{n}_{\mathrm{C}}^*/n_{\mathrm{C}}^*$ shown in Figures 8 and 9 by the relation: $\delta_{[n_{\mathrm{C}}^*]} = 1 - \hat{n}_{\mathrm{C}}^*/n_{\mathrm{C}}^*$. The resulting expression $\hat{n}_{\mathrm{C}}^* = n_{\mathrm{C}}^*(1 + \delta_{[n_{\mathrm{C}}^*]})$ is then used to anticipate relative errors on the actual atomic ratios and OM/OC ratio as follows:

$$\delta_{[n_a^*/n_{\mathrm{C}}^*]} = 1 - \frac{[n_a^*/n_{\mathrm{C}}^*]/\left(1 + \delta_{[n_{\mathrm{C}}^*]}\right)}{[n_a^*/n_{\mathrm{C}}^*]} = 1 - \frac{1}{1 + \delta_{[n_{\mathrm{C}}^*]}} \tag{D1}$$

10  $$\delta_{[OM/OC]} = 1 - \frac{1 + \left([OM/OC] - 1\right)/\left(1 + \delta_{[n_{\mathrm{C}}^*]}\right)}{[OM/OC]} = 1 - \left(\frac{1}{1 + \delta_{[n_{\mathrm{C}}^*]}} + \frac{1}{[OM/OC]} - \frac{1}{[OM/OC]\left(1 + \delta_{[n_{\mathrm{C}}^*]}\right)}\right) \tag{D2}$$

*Author contributions.* S. Takahama and G. Ruggeri designed and performed the analysis. S. Takahama wrote the manuscript.

*Acknowledgements.* Funding was provided by the Swiss National Science Foundation (200021_143298).





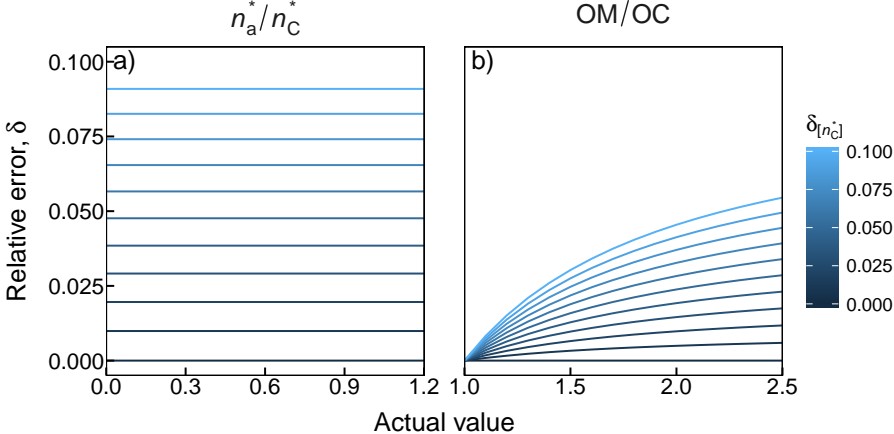

**Figure D1.** Magnitude of relative errors in atomic ratios ($\delta_{[n_a^*/n_C^*]}$) and OM/OC mass ratios ($\delta_{[OM/OC]}$) due to relative errors ($\delta_{[n_C^*]}$) in the estimation of number of carbon atoms $n_C^*$. Ten colored lines shown in each panel correspond to values of $\delta_{[n_C^*]} = \{0.0, 0.01, 0.02, \ldots, 0.1\}$.

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





## Tables

**Table 1.** Average number of atoms attached to each type of bond assumed for various types of mixtures. $\lambda_{C,COH} = \lambda_{C,carbonyl} = 1$.

| Study | Mixture type | $\lambda_{C,CH}$ | $\lambda_{C,COH}$ | $\lambda_{C,CONO2}$ |
|---|---|---|---|---|
| Allen et al. (1994) | ambient | 0.5 | | 1 |
| Russell (2003) | ambient | 0.5 | 1 | |
| Reff et al. (2007) | indoor/ambient | 0.48 | | |
| Chhabra et al. (2011) | $\alpha$-pinene SOA | 0.63 | 0.63 | 0.63 |
| | guaiacol SOA | 0.88 | 0.88 | 0.88 |
| Several[*] | ambient | 0.5 | 0.5 | 0.25 |
| Ruthenburg et al. (2014) | ambient | 0.5 | 0 | |

[*] reflects assumptions by Russell et al. (2009), Liu et al. (2009), and Day et al. (2010).

**Table 2.** Values for $\lambda_C$ with standard errors in parentheses where available (uncertainties were not calculated for the constrained optimization algorithm in the MIXTURE estimation method). Values for $\lambda_{C,COH} = \lambda_{C,carbonyl} = 1$ are fixed and therefore not included in the table.

| Set | Method | aCH | aCOH | CONO$_2$ | eCH | hydroperoxide |
|---|---|---|---|---|---|---|
| Set1 | COUNT | 0.39 (0.04) | 0.52 (0.17) | 0.52 (0.17) | | |
| Set1 | COMPOUND | 0.47 (0.01) | 0.31 (0.06) | 0.64 (0.11) | | |
| Set1 | MIXTURE | 0.45 | 0.09 | 1.00 | | |
| Set1 | NOMINAL | 0.45 | 0.50 | 0.50 | | |
| Set2 | COUNT | 0.39 (0.04) | 0.52 (0.17) | 0.52 (0.17) | 0.75 (0.25) | 0.52 (0.17) |
| Set2 | COMPOUND | 0.48 (0.01) | 0.26 (0.05) | 0.54 (0.09) | 1.08 (0.20) | 0.35 (0.07) |
| Set2 | MIXTURE | 0.50 | 0.16 | 0.41 | 1.00 | 0.00 |
| Set2 | NOMINAL | 0.45 | 0.50 | 0.50 | 1.00 | 0.50 |




**Figures**

$$Y\Theta = X \qquad \text{FG balance}$$
$$Y = X\Phi \qquad \text{carbon type balance}$$
$$Y\Theta\Lambda^T = X\Lambda^T = N_{atom} \qquad \text{atom balance}$$
$$Y\zeta = Xz \qquad \text{oxidation state balance}$$

ethane:

$$H_3C \longrightarrow CH_3$$

ethanol:

$$H_3C \diagup\diagdown OH$$

$X =$

| | CH | COH |
|---|---|---|
| ethane | 6 | 0 |
| ethanol | 5 | 1 |

$Y =$

| | CH$_3$ | H$_2$COH |
|---|---|---|
| ethane | 2 | 0 |
| ethanol | 1 | 1 |

$N_{atom} =$

| | C | H | O |
|---|---|---|---|
| ethane | 2 | 6 | 0 |
| ethanol | 2 | 5 | 1 |

$z =$

| | OS |
|---|---|
| CH | -1 |
| COH | +1 |

$\Theta =$

| | CH | COH |
|---|---|---|
| CH$_3$ | 3 | 0 |
| H$_2$COH | 2 | 1 |

$\Phi =$

| | CH$_3$ | H$_2$COH |
|---|---|---|
| CH | 1/3 | 0 |
| COH | -1/6 | 1 |

$\Lambda^T =$

| | C | H | O |
|---|---|---|---|
| CH | 1/3 | 1 | 0 |
| COH | 1/3 | 1 | 1 |

$\zeta =$

| | OS$_C$ |
|---|---|
| CH$_3$ | -3 |
| H$_2$COH | -1 |

**Figure 1.** Illustration of carbon type and FG relationships for ethane and ethanol. The FG composition matrix ($X$), carbon type matrix ($Y$), and atom composition matrix ($A$) describe properties of the compounds, and the remaining arrays — oxidation state contribution vector ($z$), carbon-FG matrix ($\Theta$), FG-carbon matrix ($\Phi$), atom-FG matrix ($\Lambda$), and carbon oxidation state vector ($\zeta$) — establish their inter-relationships.





**Figure 2.** Visualization of the carbon type matrix $\Theta$ for the APIN mechanism. Radical groups are denoted with (*). Carbon types and FGs are ordered by their aerosol abundance (in decreasing order) in the APIN simulation at $t_{\max \text{SOA}}$ (Section 2.1) with each value of $OS_C$ and $z$, respectively. The numeric label for carbon types indicates the overall rank (without regard for its $OS_C$) in the APIN simulation at $t_{\max \text{SOA}}$. Formaldehyde and formic acid are subclasses of aldehyde and COOH, respectively, but are defined separately to fulfill the conditions described in Appendix S1. Further details regarding the FG definitions are provided by Ruggeri and Takahama (2016). FGs belonging to measured subset $\mathcal{J}^* = $ Set1 (Section 2.3) is colored in red; additional FGs belonging to Set2 and Full are colored in blue and green, respectively. Corresponding carbon atoms $\mathcal{C}^*$ that are associated with (i.e., detectable by) with $\mathcal{J}^*$ are shown in the same colors.




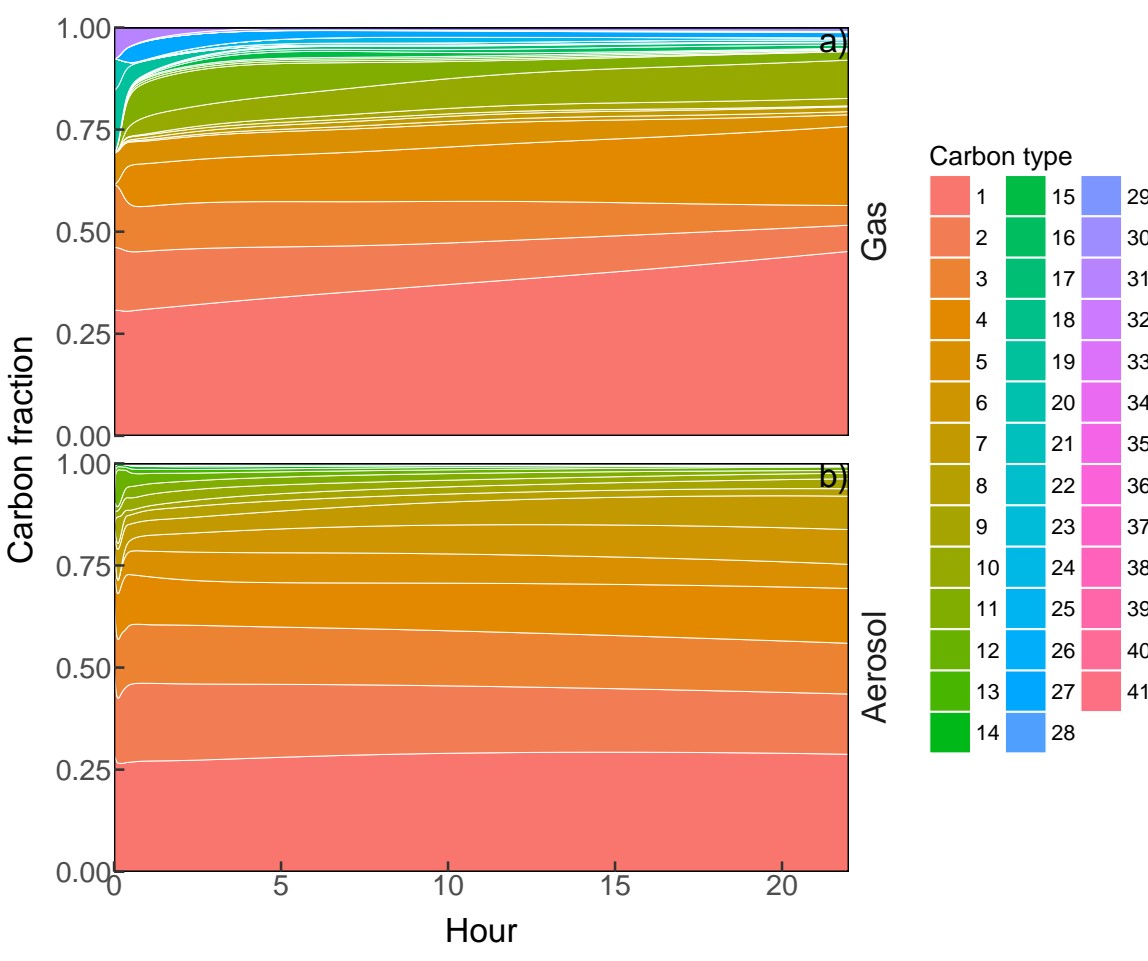

**Figure 3.** Time series of carbon type abundances for the APIN simulation described in Section 2.1. The carbon types are defined in Figure 2.





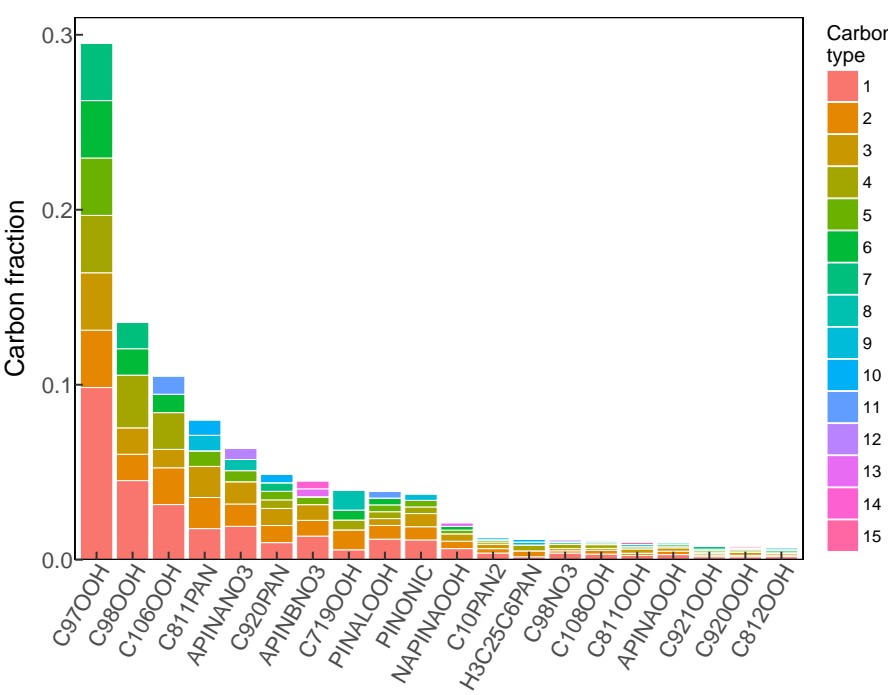

**Figure 4.** Compound and carbon type abundance for APIN simulation at $t_{\max\,\mathrm{SOA}}$. C97OOH and C98OOH are large, polyfunctional compounds containing ketone and hydroperoxide groups. The carbon types are defined in Figure 2.

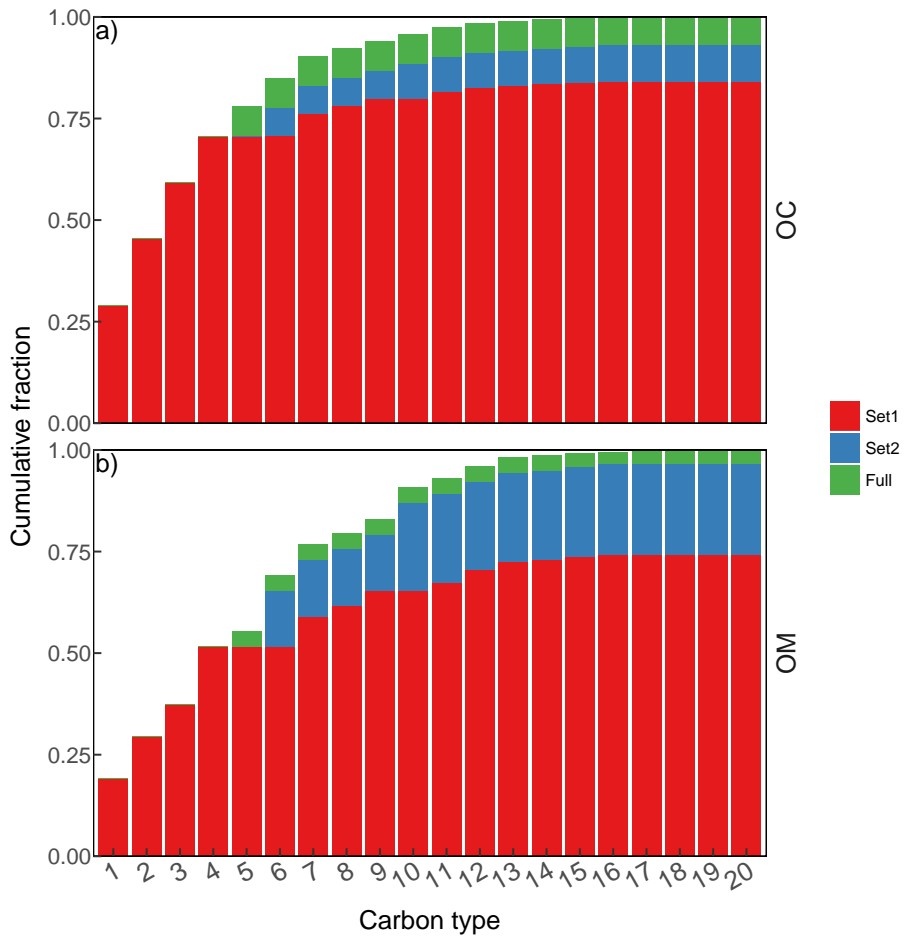

**Figure 5.** Cumulative carbon fraction for APIN simulation at $t_{\mathrm{max\,SOA}}$. Colors show carbon atoms measurable by different sets of FGs (Section 2.3). The carbon types are defined in Figure 2.





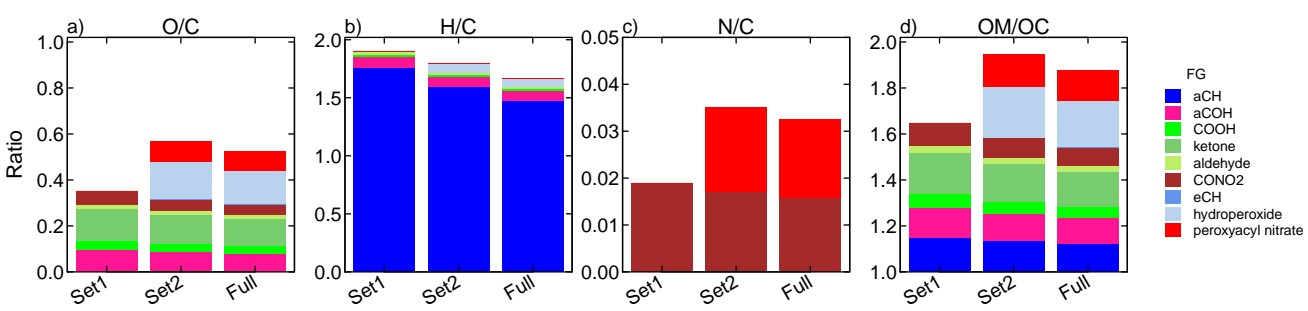

**Figure 6.** SOA properties for APIN simulation at $t_{\max \mathrm{SOA}}$. Atomic ratios ($n_a^*/n_C^*$) shown in panels (a)–(c) are in molar units, and OM/OC ratios shown in panel (d) are in mass units. The abundance of carbon used for normalization is defined by the detectable carbon for each set of FGs (Section 2.3), which can lead to estimated ratios with Set1 or Set2 to exceed the Full.





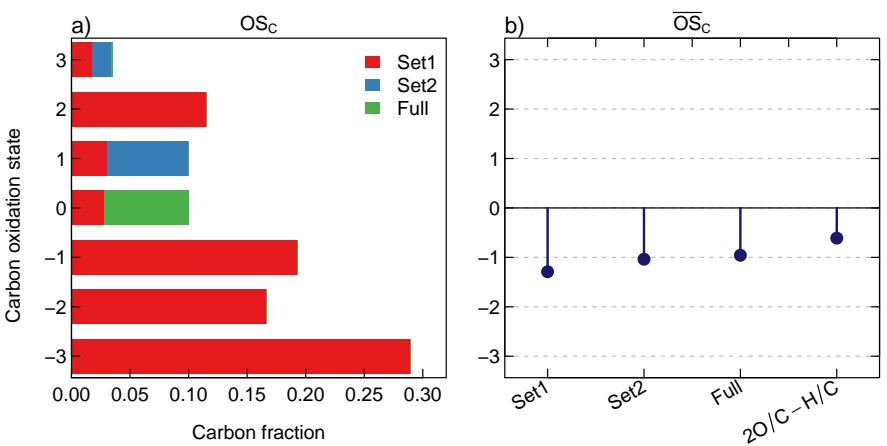

**Figure 7.** Distribution of carbon oxidation states and their ensemble estimate APIN simulation at $t_{\max \text{SOA}}$. Panel a) shows distribution and measurable carbon atoms with same color scheme 5. Panel b) shows various estimates of $OS_C$ (b) for the mixture using different FG sets (Section 2.3). $2O/C - H/C$ is a common approximation used by elemental analysis and is included for reference.





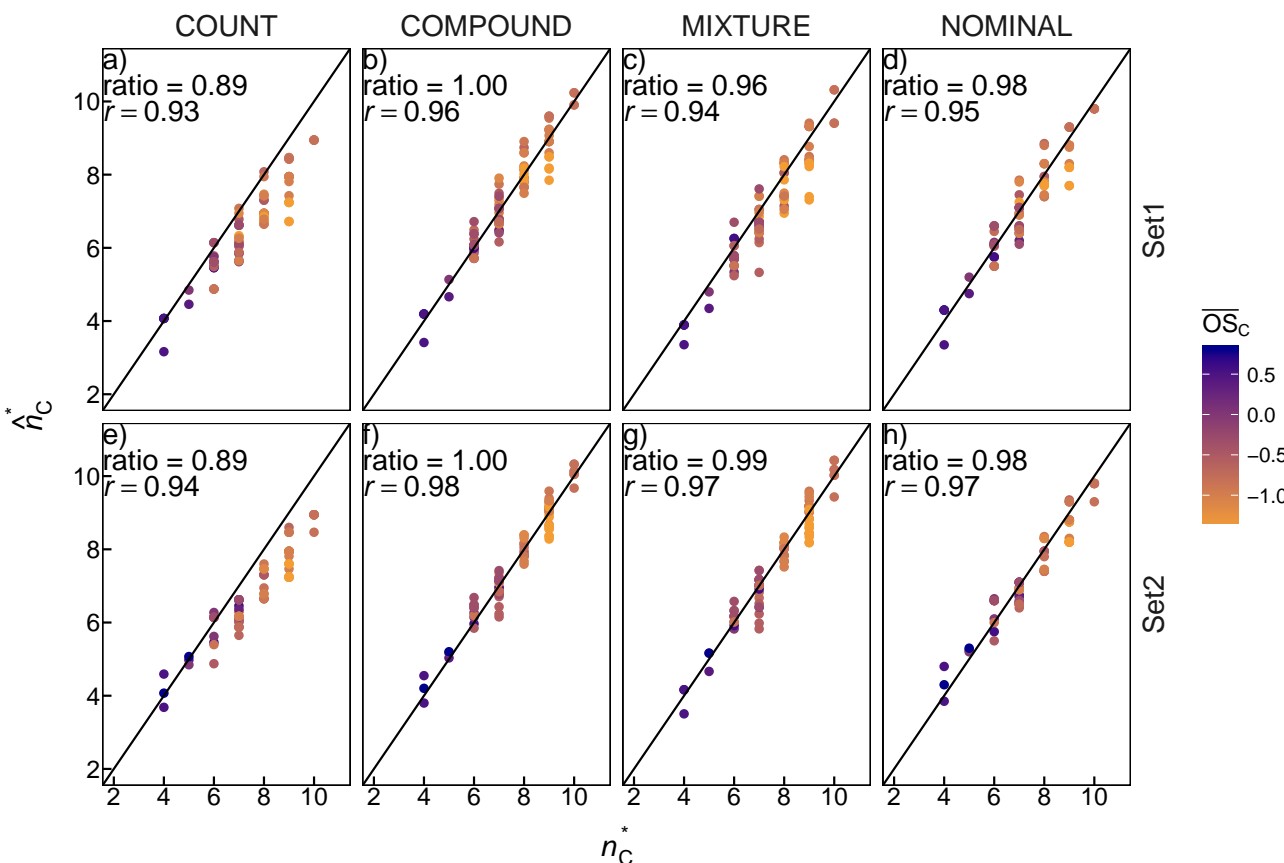

**Figure 8.** Comparison of estimated ($\hat{n}_C^*$) and actual ($n_C^*$) number of measurable carbon atoms in different SVOC compounds (colored by their compound-averaged oxidation states, $\overline{OS}_C$) using estimates of $\hat{\lambda}_C$ for various FG sets and solution methods. The diagonal line is the $x = y$ line provided for visual reference. The ratio is defined as $\hat{n}_C^*/n_C^*$ and estimated as the slope (not drawn) of $\hat{n}_C^*$ regressed on $n_C^*$. $r$ is the Pearson's correlation coefficient.





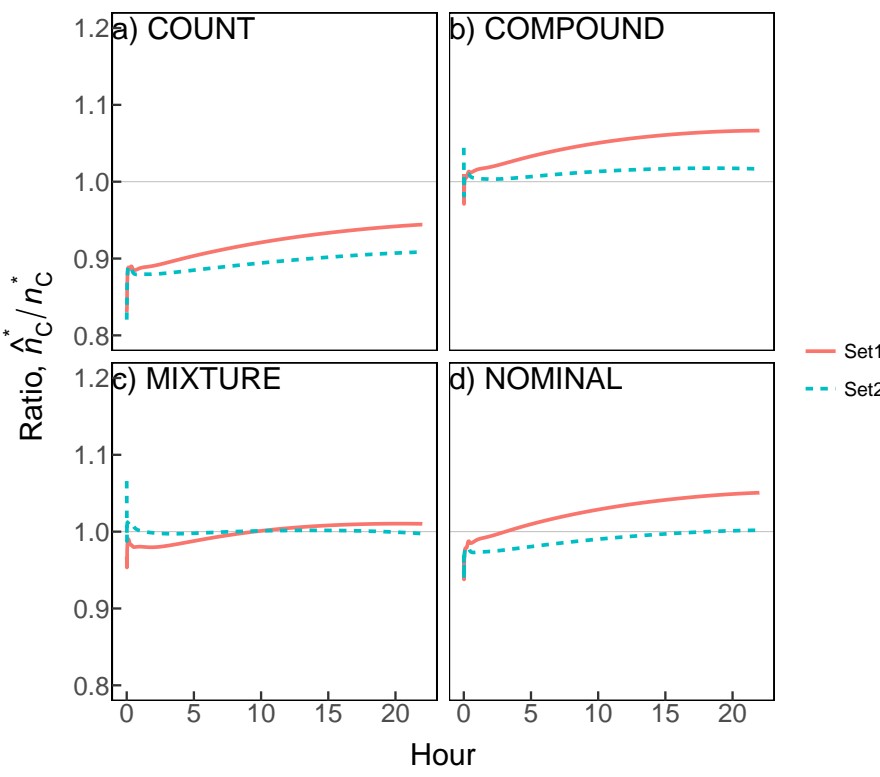

**Figure 9.** Ratios of estimated ($\hat{n}_C^*$) and actual ($n_C^*$) number of measurable carbon atoms in the APIN simulated aerosol mixture using estimates of $\hat{\lambda}_C$ for various FG sets and solution methods. The gray horizontal line corresponds to $y = 1.0$ (perfect estimate).





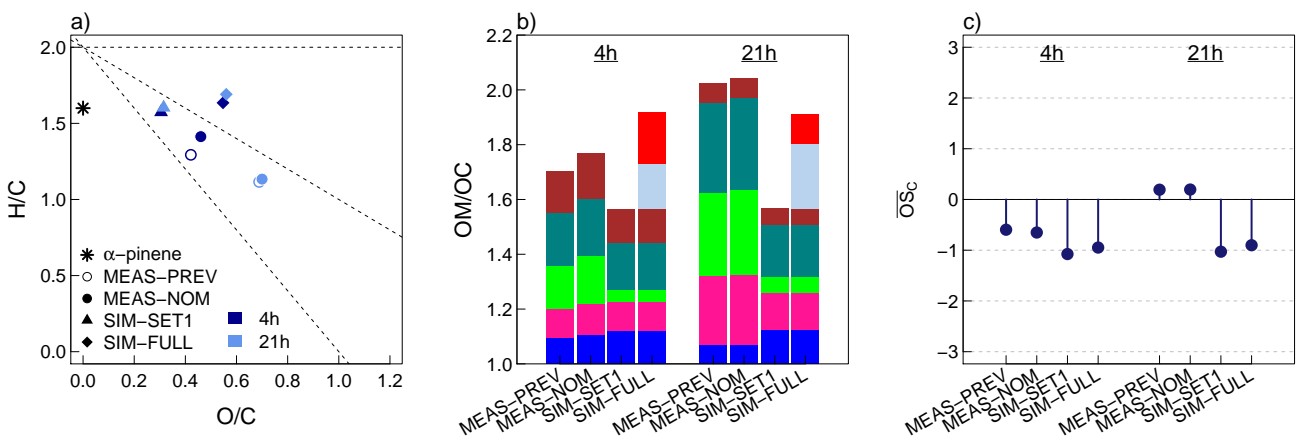

**Figure 10.** Comparison of measurement (MEAS) and simulations (SIM) for samples ending approximately at 4 and 21 hours (time-integrated over 3.1 to 4.2 hours and 17.6 and 21.6 hours, respectively) after initiation of photochemistry (Sax et al., 2005; Ruggeri et al., 2016). Further details on labels for estimates are defined in Section 3.4. Colors for (b) are the same as for Figure 6, except that ketone and aldehyde has been combined into a single color (teal) because the reported measurements do not differentiate between the two types of carbonyl.