# Peer review of "Supporting Information for Technical Note: Relating functional group measurements to carbon types for improved model-measurement comparisons of organic aerosol composition"

_Atmospheric Chemistry and Physics, 2016_

## Referee Comment (RC1) · Anonymous Referee #1 · 25 Nov 2016

General Comments

In this paper the authors report further on a framework they have been developing to extend the utility of functional group analysis of organic aerosol to obtain more accurate information on possible missing chemical components, organic mass/organic carbon ratio, elemental composition (O/C ratio), and carbon oxidation state. A thorough discussion of the conceptual and analytical aspects of the methods are presented and discussed, and software is made available that can be downloaded by others wishing to use these methods. The methods were applied to a data set of SOA composition

predicted for a-pinene photooxidation using the MCM coupled with gas-particle partitioning, and for which FTIR functional group data were also available. A variety of useful results are extracted from the analysis. The paper is technically very dense and beyond my ability to fully understand without a much greater investment of time and effort, and so I am unable to evaluate many of the details that are presented. Nonetheless, the approach seems reasonable to me.

Overall, I think this is likely to be a useful paper as more people begin to adapt these methods for analysis of functional group data. Not many group conduct functional group analysis, but there are reasons to think this could increase in the future because of its unique value compared with molecular and AMS analysis. I'm very pleased to see that the authors have made the software needed to conduct the analysis freely available. I recommend the paper be published after these very minor comments are addressed.

Specific Comments

Page 5, line 7: I have never heard it said that FTIR measures the total abundance of bonds. The spectrum depends on vibrations and bending of molecules, but I don't see how this translates directly into bond abundance. This should be clarified.

Page 13, lines 4–5: Baltensperger and co-workers (AMT) measured a peroxide lifetime of a few hours in chamber SOA generated from a-pinene + ozone.

Technical Comments

Page 1, line 17: Insert "a" before "framework".

Page 3, line 1: Should "metric" be inserted after "carbon-centric"?

Page 4, line 8: Delete "a" after "specific".

Page 5, line 10 or 11. Delete "weighted" on one line or the other.

Page 6, line 1: Can probably delete "expressed".

Page 6, line 2: Should be "below" not "blow".

Page 7, line 4: Should "atoms" be inserted after "carbon"?

Page 7, line 11: Should a comma be added after "eCH"?

---

## Referee Comment (RC2) · Anonymous Referee #2 · 9 Jan 2017

The present paper is a technical note to already-published model-measurement comparison SOA studies (Ruggeri et al., ACP 2016). The general scientific objectives are already explained in the previous publications. Specifically, the Authors aim to exploit organic functional group distributions to constrain explicit-chemistry models of secondary organic aerosol (SOA) formation. The idea, although not new, is sound, because functional groups provide direct information about organic reactivity (like in the Carbon Bond Mechanism developed long ago for gas-phase reactions) and keep track of the chemical mechanisms that govern the enrichment of SVOC in the particulate phase. This technical note, in particular, focuses on the derivation of carbon-based

metrics (such as oxidation state, and O/C ratios) from measurable functional groups distributions, with the aim to support and inform the comparison between FG-based techniques (such as FTIR) and more established mass spectrometric methods (AMS). The methodology is discussed in detail, and for the first time O/C ratios from FTIR measurements are reported taking into account the possible biases due to the selectivity of FTIR spectroscopy for specific FGs. I have only specific comments:

1. Simple examples, like the one shown in Figure 1, are essential for a chemist who is not familiar with linear algebra. I invite the Authors to comment such examples in the text, or in the Supplementary Information. For instance, it is not straightforward why negative values for phi can be obtained. This is important also to understand why values for lambda lower than 1/3 (the theoretical value for a tri-substituted carbon atom) are found in Table 2.

2. The three methods for carbon abundance estimation discussed in Section 2.4 are compared for the examples of SOA formation considered in this study (Table 2, Figures 8, 9), but it is difficult to derive general conclusions on their applicability. How much details on the qualitative composition (in "carbon types") must be known a priori? It would be important to expand the paragraph about the type and nature of the datasets which the three methods rely on.

3. The paper makes use of the notion of "carbon types". These are exemplified for simple molecules in Figure 1, and are otherwise listed as numbers in the other Tables and Figures. I suggest to explicit the full list in the Supplementary Information. It will be important to understand how many carbon types contain heteroatoms in functional groups (alcohols, carboxylic acids, nitro groups etc.), which seems to be the focus of the paper, instead of being included in the skeleton of the molecules (ethers, esters, etc.).
* * *

---

## Author Response (AR1)

**Response to Reviewer #1**

General comments:

In this paper the authors report further on a framework they have been developing to extend the utility of functional group analysis of organic aerosol to obtain more accurate information on possible missing chemical components, organic mass/organic carbon ratio, elemental composition (O/C ratio), and carbon oxidation state. A thorough discussion of the conceptual and analytical aspects of the methods are presented and discussed, and software is made available that can be downloaded by others wishing to use these methods. The methods were applied to a data set of SOA composition predicted for a-pinene photooxidation using the MCM coupled with gas-particle partitioning, and for which FTIR functional group data were also available. A variety of useful results are extracted from the analysis. The paper is technically very dense and beyond my ability to fully understand without a much greater investment of time and effort, and so I am unable to evaluate many of the details that are presented. Nonetheless, the approach seems reasonable to me.

Overall, I think this is likely to be a useful paper as more people begin to adapt these methods for analysis of functional group data. Not many group conduct functional group analysis, but there are reasons to think this could increase in the future because of its unique value compared with molecular and AMS analysis. Im very pleased to see that the authors have made the software needed to conduct the analysis freely available. I recommend the paper be published after these very minor comments are addressed.

**We thank the reviewer for this encouraging assessment.**

Specific comments:

1. Page 5, line 7: I have never heard it said that FTIR measures the total abundance of bonds. The spectrum depends on vibrations and bending of molecules, but I dont see how this translates directly into bond abundance. This should be clarified.

The absorbances associated with stretching and bending modes of molecular bonds can be calibrated (according to Beer-Lambert law) to quantify their abundances, but it may have been confusing to include the word, "total", which was meant to indicate the independence from extraction or ionization efficiencies often used in other types of chemical analysis. Also, describing the method of quantification is not the main purpose of this Section 2.2. To remedy these issues, we have first simplified the expression to in Section 2.2:

"The molar abundance of molecules  $\mathbf{n}_{\text{molec}} = [n_i]$  in a mixture (consisting of a set of molecules denoted by  $\mathcal{M}$ ) can be related to FG abundance  $\mathbf{n}_{\text{group}} = [n_j]$  (for each FG in  $\mathcal{J}$ ) obtained by FT-IR — or other means — by invoking a group composition matrix  $\mathbf{X} = [x_{ij}]$ , which describes the FG makeup of each molecule."

We have also included the appropriate references for obtaining bond or FG abundances from FT-IR spectra and other methods in Section 1:

"However, studies on this topic have thus far been very limited on account of challenges in quantitative characterization of FGs, which requires either advanced algorithms (e.g., Takahama et al., 2013; Ruthenburg et al., 2014; Takahama and Dillner, 2015) for spectral interpretation or derivitization steps (e.g., Dron et al., 2010; Aimanant and Ziemann, 2013) for chemical analysis."

Furthermore, we have edited Section 1 to make note of the fact that we are not considering less than "total recovery" of bonds as we do not chemically extract our samples:

"The benefit of developing a systematic approach is that we can precisely understand the achievable mass recovery, and biases incurred on the calculated O/C and OM/OC for a given set of molecules and FGs analyzed (when chemical extraction is not required, OM mass recovery is primarily dependent on the completeness of FG calibration models constructed)."

2. Page 13, lines 4–5: Baltensperger and co-workers (AMT) measured a peroxide lifetime of a few hours in chamber SOA generated from a-pinene + ozone.

We thank the reviewer for pointing out this important reference (in Cell) — the text has been modified to include this citation:

"The rate of transformation of these FGs remains uncertain — for instance, reported lifetimes of hydroperoxides range from less than an hour to many days (Epstein et al., 2014; Krapf et al., 2016); resolving their reaction pathways may play a critical role in understanding model-measurement discrepancies (McVay et al., 2016)."

Technical comments:

- 1. Page 1, line 17: Insert "a" before "framework".
- 2. Page 3, line 1: Should "metric" be inserted after "carbon-centric"?
- 3. Page 4, line 8: Delete "a" after "specific".
- 4. Page 5, line 10 or 11. Delete "weighted" on one line or the other. Page 6, line 1: Can probably delete "expressed".
- 5. Page 6, line 2: Should be "below" not "blow".
- 6. Page 7, line 4: Should "atoms" be inserted after "carbon"?
- 7. Page 7, line 11: Should a comma be added after "eCH"?

We thank the reviewer for these technical corrections — the changes have been made in the manuscript.

**Response to Reviewer #2**

The present paper is a technical note to already-published model-measurement comparison SOA studies (Ruggeri et al., ACP 2016). The general scientific objectives are already explained in the previous publications. Specifically, the Authors aim to exploit organic functional group distributions to constrain explicit-chemistry models of secondary organic aerosol (SOA) formation. The idea, although not new, is sound, because functional groups provide direct information about organic reactivity (like in the Carbon Bond Mechanism developed long ago for gas-phase reactions) and keep track of the chemical mechanisms that govern the enrichment of SVOC in the particulate phase. This technical note, in particular, focuses on the derivation of carbon-based metrics (such as oxidation state, and O/C ratios) from measurable functional groups distributions, with the aim to support and inform the comparison between FG-based techniques (such as FTIR) and more established mass spectrometric methods (AMS). The methodology is discussed in detail, and for the first time O/C ratios from FTIR measurements are reported taking into account the possible biases due to the selectivity of FTIR spectroscopy for specific FGs.

We thank the reviewer for this lucid assessment.

Specific comments:

1. Simple examples, like the one shown in Figure 1, are essential for a chemist who is not familiar with linear algebra. I invite the Authors to comment such examples in the text, or in the Supplementary Information. For instance, it is not straightforward why negative values for phi can be obtained. This is important also to understand why values for lambda lower than 1/3 (the theoretical value for a tri-substituted carbon atom) are found in Table 2.

As the author correctly notes, lambda values should nominally take on values rational values of  $\{1/4, 1/3, 1/2, 1\}$  at the level of individual carbon atoms. However, the coefficients reported represent a single set of values to be used across all carbon types, which leads to irrational values. A value of lambda below 1/3 results from the empirical nature of regression methods, in which obtained coefficient values are insensitive to under-represented FGs (therefore could be replaced with 1/3 with little impact on results), or eliminated (set to zero) in the case of redundant FGs (i.e. strongly correlated to another FG). We have included the following statements in Section 2.4 to address potential questions from other readers:

"Each of the solutions produces a series of irrational numbers (due to the multiplicitous configurations of FGs on carbon atoms) that may be overly precise for the data set used for estimation."

and

"Direct fitting methods, on the other hand, may lead to insignificant coefficients from under-represented or redundant FGs [...]."

While lambda may have a physical interpretation in simple instances (not in complex mixtures with multiplicitous configurations of functional groups as noted above), negative

values for phi are permitted in that they are only required to satisfy the carbon type balance (equation 4). We have added the following statement to Section 2.4 to better connect readers with the illustration in Figure 1:

"The elements of  $\Phi$  satisfy the carbon type balance (equation 4) but are not required to be non-negative, but their summation across rows (equation 6) yields values for  $\lambda_{\rm C}$  that corresponds to the number of carbon atoms per FG associated with them.

[...]

" $\lambda_{\rm C}$  may also not correspond to a physically interpretable quantity in such instances, as a single set of coefficients are insufficient to estimate the exact abundances of carbon atoms under these circumstances."

2. The three methods for carbon abundance estimation discussed in Section 2.4 are compared for the examples of SOA formation considered in this study (Table 2, Figures 8, 9), but it is difficult to derive general conclusions on their applicability. How much details on the qualitative composition (in carbon types) must be known a priori? It would be important to expand the paragraph about the type and nature of the datasets which the three methods rely on.

It is yet difficult to recommend a best method for estimation in a general sense, but we hope that this exploratory study can initiate further inquiry into this topic. However, as noted by the reviewer, it is worth considering the nature of underlying data sets in more detail to describe their tradeoffs. We had previously written in Section 2.4:

"Numerical details aside, the main differences among the three are the data sets used for estimation. COUNT uses information from  $\Theta$  only (defined for the FGs in the APIN mechanism), COMPOUND uses carbon type abundances in compounds (limited to SVOCs in the APIN mechanism), and MIXTURE uses mixture information of the condensedphase (from different periods in the APIN simulation)."

However, the important difference which may have not been emphasized was the weighting of the estimates. We have added to Section 2.4 the following explanation:

"The resulting differences in estimates of  $\hat{\lambda}_{\rm C}$  are largely due to weighting of FGs associated with each carbon type: each type receiving equal weight (COUNT), by frequency of occurrence in SVOCs (COMPOUND), and by abundance in SOA formed in the APIN simulation (MIXTURE). While the COUNT method is physically significant at the level of individual carbon atoms, the representativeness of estimated values for use in mixtures can vary according to composition. Direct fitting methods, on the other hand, may suffer from errant coefficients from redundant or under-represented FGs, or be overly specific such that they cannot be generalized to other systems. Therefore, the results from all three methods are evaluated to explore the range of plausible values."

Regarding the COUNT method, we have also added the following statement to aid the physical significance of this estimate:

"The main premise of this approach is to apportion fractional units of carbon to each measured FG such that their sum equals unity."

3. The paper makes use of the notion of carbon types. These are exemplified for simple molecules in Figure 1, and are otherwise listed as numbers in the other Tables and Figures. I suggest to explicit the full list in the Supplementary Information. It will be important to

understand how many carbon types contain heteroatoms in functional groups (alcohols, carboxylic acids, nitro groups etc.), which seems to be the focus of the paper, instead of being included in the skeleton of the molecules (ethers, esters, etc.).

We thank the reviewer for this comment. We had discussed the case of anhydrides, esters, and organic peroxides in the supporting information but had not referred to it in the main text (ethers are currently out of the scope of our analysis as they are not included in mechanisms which we have studied). We have modified Section 2.3 to include the following statements:

"All elements in equation 3 can be known precisely for any set of molecules  $\mathcal{M}$  from the chemometric patterns and atom-level validation described by Ruggeri and Takahama (2016), and is summarized in Section S1. Furthermore, the FGs included in the APIN system are all those which are defined by association only to single carbon atoms (e.g., alcohol, carboxylic, methylene groups). Methods for extending this analysis to FGs containing multiple carbon atoms (e.g., anhydride, ester, and organic peroxide groups) are described in Section S2."

While the APIN system does not include these multi-carbon functional groups, we have included additional analyses (Supporting Information, Section S2, and Tables S1–S4, and online code repository) to conclude that at least in terms of frequency of occurrence in  $\alpha$ pinene and 1,3,5-trimethylbenzene degradation systems studied by Ruggeri et al. (2016), our 41 carbon types presented in the main text encompasses 92% of the 2867 carbon atoms in the 441 molecules. The four tables can be viewed in the Supporing Information, but the text is copied below:

"Tables S1–S3 show carbon atom types associated with single-carbon FGs (conversely stated, each FG is uniquely associated with one carbon atom), two-carbon FGs (carbon atoms in these FGs share some heteroatoms with other carbon atoms), and carbononly structures present in the combined set of molecules from the  $\alpha$ -pinene and 1,3,5trimethylbenzene degradation schemes. In this set of 441 molecules, there are 2867 carbon atoms that can be classified into one of 60 types (labeled in order of frequency, X1–X60, prefixed by character 'X' to prevent confusion with carbon type labels used in the APIN simulation) that differ in their association with 30 unique FGs. 46 of these types contain unique FGs (2557 / 2867 carbon atoms belong in this category), 11 of these types share FGs (116 / 2867 carbon atoms belong in this category), and 3 are bonded only to other carbon atoms (194 / 2867 carbon atoms belong in this category). 92% of the carbon atoms in this superset belong to the 41 carbon types (which includes two of the tertiary and quaternary carbon types) from the APIN simulation discussed in the main body of this manuscript, though this relative abundance is reported on a frequency basis and does not consider molecular abundances that might be typical in a SOA mixture. The correspondence of labels used in the main document (numbered by abundance of total carbon during the APIN simulation) and Tables S1–S3 (numbered by frequency of occurrence of in the 441 molecules) are listed in Table S4."

The rest of Section S2 is dedicated to describing adjustments necessary to include carbon atoms associated with anhydrides, esters, and peroxides for estimation of carbon-centered metrics, and can be applied to any functional groups with which more than one carbon atom would be associated. The text has been modified to indicate that our framework is quite general in this sense (Section S2):

"The second generalization concerns FGs that contain skeletal heteroatoms. FGs of this

type — specifically in this case, anhydride, ester, and (organic) peroxide — are present in photooxidation products of 1,3,5-trimethylbenzene included in the MCMv3.2 mechanism (Bloss et al., 2005; Ruggeri et al., 2016), and corresponding SMARTS patterns were developed by Ruggeri and Takahama (2016) to match these structures. Equation S3 should accordingly permit two carbon atoms to be associated with each of these exceptional FGs. To accommodate such groups (and other FGs defined by membership of multiple carbon atoms) in our framework, the carbon type formulation can be a) extended to "carbon units" consisting of one or more carbon atoms and their bonded heteroatoms, or b) modified by the introduction of a correction factor."

[revised manuscript text omitted]
_{\rm C}^*]}$  with respect to  $n_{\rm C}^*$  such that  $\epsilon = \delta_{[n_{\rm C}^*]} n_{\rm C}^*$ . The magnitude of  $\delta_{[n_{\rm C}^*]}$  can be associated with

Table B1. Mathematical symbols used in the manuscript and their descriptions.

| Category  | Symbol                                                               | Description                                            |
|-----------|----------------------------------------------------------------------|--------------------------------------------------------|
| Indices   | i                                                                    | compound or molecule index                             |
|           | k                                                                    | carbon type index                                      |
|           | j                                                                    | FG index                                               |
|           | a                                                                    | atom index                                             |
| Variables | n                                                                    | number of moles of a substance (atom, compound, or FG) |
|           | $\mathbf{X} = [x_{ij}]$                                              | group composition matrix                               |
|           | $\mathbf{Y} = [y_{ik}]$                                              | carbon type matrix                                     |
|           | $\boldsymbol{\Theta} = [\theta_{kj}]$                                | carbon-group matrix                                    |
|           | $\mathbf{\Phi} = [\phi_{jk}]$                                        | group-carbon matrix                                    |
|           | $oldsymbol{\zeta} = [\zeta_k]$                                       | carbon type oxidation state vector                     |
|           | $oldsymbol{z} = [z_j]$                                               | oxidation state contribution vector                    |
|           | $\mathbf{\Lambda} = [\lambda_{aj}]$                                  | atom-group matrix                                      |
|           | $\boldsymbol{\lambda}_{\mathrm{C}} = [\hat{\lambda}_{\mathrm{C},j}]$ | carbon atom-group vector                               |
|           | $OS_{\mathrm{C}}$                                                    | carbon oxidation state                                 |
|           | $\overline{OS}_{\rm C}$                                              | mean carbon oxidation state                            |
| Sets      | $\mathcal{A}$                                                        | set of atoms                                           |
|           | $\mathcal{M}$                                                        | set of molecule types                                  |
|           | ${\mathcal J}$                                                       | set of FGs                                             |
|           | $\mathcal{C}$                                                        | set of carbon types                                    |

Table C1. Absorption bands in the mid-infrared for vibrational modes present in FGs proposed for Set2 (Section 2.3).

| FG                                | $\tilde{\nu}  (\mathrm{cm}^{-1})$ | description                            |
|-----------------------------------|-----------------------------------|----------------------------------------|
| eCH 1                  | 3005-2980                         | C-H stretch                            |
| $hydroperoxide^2$                 | 3300-3400                         | OO-H stretch (strong)                  |
|                                   | 860-840                           | O-OH stretch (weak)                    |
| peroxyacyl nitrate 2,3 | 760-849                           | NO scissoring                          |
|                                   | 1340-1223                         | NO 2 symmetric stretch      |
|                                   | 1777-1700                         | $\mathrm{NO}_2$ anti-symmetric stretch |
|                                   | 1880–1777                         | C=O stretch                            |

1Maria et al. (2003); 2Shurvell (2006); 3Monedero et al. (2008)

the ratio  $\hat{n}_{\rm C}^*/n_{\rm C}^*$  shown in Figures 8 and 9 by the relation:  $\delta_{[n_{\rm C}^*]} = 1 - \hat{n}_{\rm C}^*/n_{\rm C}^*$ . The resulting expression  $\hat{n}_{\rm C}^* = n_{\rm C}^*(1 + \delta_{[n_{\rm C}^*]})$  is then used to anticipate relative errors on the actual atomic ratios and OM/OC ratio as follows:

$$\delta_{[n_a^*/n_C^*]} = 1 - \frac{[n_a^*/n_C^*] / (1 + \delta_{[n_C^*]})}{[n_a^*/n_C^*]} = 1 - \frac{1}{1 + \delta_{[n_C^*]}}$$
(D1)

[revised manuscript text omitted]
  | 0.45        | 0.50        | 0.50              | 1.00        | 0.50          |